# The Effects of Particulate Matter Alert on Urban Park Visitation in Seoul, Korea: Using Segmented Regression

**DOI:** 10.3390/ijerph192215372

**Published:** 2022-11-21

**Authors:** Yongsoo Choi, Garam Byun, Jong-Tae Lee

**Affiliations:** 1School of Health Policy and Management, College of Health Science, Korea University, 145, Anam-ro, Seongbuk-gu, Seoul 02841, Republic of Korea; 2Interdisciplinary Program in Precision Public Health, Korea University, 145, Anam-ro, Seongbuk-gu, Seoul 02841, Republic of Korea

**Keywords:** air pollution alert, policy evaluation, urban park visit, particulate matter

## Abstract

To reduce the health burden from particulate matter (PM), the Korean government implemented a nationwide PM_10_ (particles less than 10 µg/m^3^ in diameter) alert system in 2015. The policy was intended to reduce PM exposure by advising people to refrain from outdoor activities on highly polluted days. The present study aimed to estimate the effect of the PM_10_ alert system on people’s daily outdoor activity patterns using urban park (specifically, Children’s Grand Park) visitation data from Seoul, South Korea, from 2014–2019. Segmented regression was fitted to estimate whether the number of visitors to the park decreased on the days with PM_10_ alerts. PM_10_ concentration of 80 µg/m^3^, the cut-off point for a “Bad” alert, was set as a threshold, and discontinuity at the threshold and change in the relative risk after the threshold was tested. Time series regression was used to estimate the dose–response line between the ambient PM_10_ concentration and the daily number of park visitors. The number of park visitors decreased by 11.8% (relative risk: 0.881, 95% confidence interval: 0.808, 0.960) when a “Bad” alert was issued (PM_10_ level above 80 µg/m^3^) compared to when the alert level was “Normal” (PM_10_ level less than 80 µg/m^3^). The present study found evidence that the PM_10_ alert influenced people’s daily outdoor activities in Seoul, Korea. As the main purpose of the PM alert is to encourage people to refrain from outdoor activities, evaluating the relationship between PM alerts and behavior patterns can help to grasp the effectiveness of the policy. Further efforts should be made to investigate whether the observed behavioral change leads to reductions in health outcomes caused by PM.

## 1. Introduction

Particulate matter (PM) refers to a mixture of particles suspended in the air that vary in size and composition. PM is categorized by its aerodynamic diameter; those with ≤10 µm are categorized as PM_10,_ and those ≤2.5 µm are categorized as PM_2.5_. According to the World Health Organization (WHO), PM_2.5_ is the fifth leading risk factor for death worldwide, causing 4.2 million mortalities in 2015 [1]. Besides mortality, PM has been reported to increase hospitalization rates for various diseases, including cardiovascular and respiratory diseases [2,3].

Many countries or governments have made numerous efforts to reduce the health effects of PM, and one of the most popular options was to adopt a PM alert system. The alert system aims to reduce personal PM exposure by predicting the ambient concentration in advance and warning people to take avoidance behaviors when episodic spikes in PM concentration are expected. In the case of the Republic of Korea (hereafter Korea), the local government of Seoul first introduced a PM_10_ alert system in 2005. However, the early system had several limitations. Public awareness of PM was low then, and people were not sufficiently motivated to change their behaviors. According to a survey conducted in 2011, only 16% of people responded to the alert system by taking avoidance behaviors [4]. 

In 2013, as the WHO designated the outdoor air pollutant as a Group 1 carcinogen, public concerns about PM grew sharply in Korea [5]. To respond to the growing concerns, the Korean Ministry of Environment implemented a nationwide PM_10_ alert system in 2014, replacing the existing alert system operated by the Seoul Government. Although the new system had the same standard as the previous one, it had some improvements. It published alerts more actively through various channels, including the internet, television, short message service, and smartphone apps [6]. The proportion of respondents who take avoidance behaviors when referring to the alert system increased from 16% to 32% in the year the nationwide alert system was implemented [4]. In 2018, 738 out of 1111 people surveyed (66%) answered that they had searched the alerts to take precautionary measures [7]. 

To make a policy more sustainable and efficient, it is important to evaluate its effectiveness. In the case of Korea, however, few studies have estimated the effects of the PM alert so far. As the primary purpose of the policy is to trigger behavioral changes in people, it is important to identify whether people refrain from going outside when the alerts are issued. To the authors’ best knowledge, one study evaluated the effectiveness of the alert system with reference to the number of visitors to tourist attractions such as the national museum, but found no associations [8]. 

The present study aimed to estimate the effects of the nationwide PM_10_ alert system on a more regular daily outdoor activity, visiting an urban park. Segmented regression analysis was used to estimate the association between the alert system and the number of visitors to the Seoul Children’s Grand Park located in Seoul, Korea, from 2014–2019.

## 2. Materials and Methods

### 2.1. Study Population and Data Sources

Figure 1 describes the history of PM_10_ alert systems in Seoul, Korea. As mentioned in the introduction, the Seoul government first introduced a local PM_10_ alert system in 2005, and it was replaced by the nationwide alert system operated by the Ministry of Environment in 2014. The primary purpose of this study was to evaluate the effectiveness of the nationwide alert system from 7 February 2014 to 31 December 2019 (“Study period” in Figure 1). In addition, as a temporal control for the main analysis, the effects of policy for the five years before the introduction of the nationwide alert system were also analyzed (“Control period” in Figure 1).

The criteria and instructions of the nationwide PM_10_ alert system are shown in Appendix A. The Ministry of Environment predicts PM_10_ concentrations up to two days ahead, and a “Bad” alert is issued if a daily predicted concentration exceeds 80 µg/m^3^. The predicted result is published through various channels four times a day (5 A.M., 11 A.M., 5 P.M., and 11 P.M.), and people are advised to stay indoors when the alert is issued.

The daily number of visitors to the Seoul Children’s Grand Park was used to measure the outdoor activity pattern of the Seoul population. The park is located in the east of Seoul, with an area of 536,088 m^2^. Over half of the park is covered with green spaces (60.3%), and facilities such as zoos and botanical gardens take up the rest of the area (39.7%). It first opened in 1973 and became free of charge in 2006, allowing people to freely use it during opening hours (05:00 to 22:00). Visitors can enter the park only through designated entrances, and automated visitor counting machines installed at all entrances measure the number of visitors to the park. The park entrance data are open to the public and can be downloaded from the Seoul Facility Corporation website (https://www.data.go.kr/en/index.do, accessed on 20 November 2022).

Ambient PM_10_ concentrations measured by the National Institute of Environmental Research (NIER) were obtained from their repository (https://www.airkorea.or.kr/eng/, accessed on 20 November 2022). Hourly measured PM_10_ data from 25 measurement stations were averaged to calculate the daily representative PM_10_ concentration of Seoul. Categorical alert results that were issued for December 2014–April 2019 were also obtained from the NIER. The data contains the history of the alert results in one of four categories (“Good”, “Normal”, “Bad”, and “Very bad”). The data were re-categorized into two levels (“Good” or “Normal” and “Bad” or “Very bad”) and used for sensitivity analysis. Daily meteorological data, including temperature, relative humidity, precipitation, and snowfall, were obtained from the Korea Meteorological Administration website (https://data.kma.go.kr/resources/html/en/aowdp.html, accessed on 20 November 2022). 

### 2.2. Statistical Analysis

Segmented regression was fitted to estimate the effects of the PM_10_ alerts on the number of visitors to the park by setting 80 µg/m^3^ (criteria for the “Bad” alert) as a threshold [9]. A “Bad” alert was issued based on the threshold for the predicted PM_10_ concentrations, which the Ministry of Environment calculates. A discontinuity in park visitors at 80 µg/m^3^ would suggest an effect of alerts on people refraining from outdoor activities, whereas continuity would reflect a null effect. In addition, if people react more sensitively on days with more severe alerts, the policy effects would appear as a reduction in the slope (i.e., relative risk (RR)) in the dose-response line only after the threshold. Hereafter, the discontinuity effect around the threshold will be referred to as “intercept change” and the reduction in slope after the threshold will be referred to as “slope change”. Because historical forecast records of the prediction model were unavailable, ambient PM_10_ concentrations were used as a surrogate. The dose-response line between the ambient PM concentration and the number of park visitors was estimated using time series regression [10]. The statistical model for the segmented regression is as follows [9].
$$\begin{array}{cc} g\left[E\left(Y_{i}\right)\right]=\beta_{0}+ & \beta_{1}\left(PM_{i}-c\right)+D_{1}\left(P_{i}\right)+\beta_{2}\left(PM_{i}-c\right)\left(P_{i}\right)+\beta_{3}\left(RH_{i}\right)\\ & +ns\left(Time_{i},\;df=6/year\right)+D_{2}\left(Holiday_{i}\right)+D_{3}\left(DOW_{i}\right)\\ & +D_{4}\left(Event_{i}\right)+D_{5}\left(Rain_{i}\right)+D_{6}\left(Snow_{i}\right)+offset(\log\left(Population\right)) \end{array}$$

g[.] is a generalized linear function with a quasipoisson link. $Y_{i}$ denotes the number of park visitors on day i. $PM_{i}$ is the daily PM_10_ concentration and c is the cut-off value 80 µg/m^3^. $P_{i}$ is an indicator variable representing non-alert days ($P_{i}=0)$ and alert days ($P_{i}=1$). $D_{i}$ represents the effects of indicator variables. In the model, $D_{1}$ represents the intercept effect, and $\beta_{2}$ describes the slope effect of the alert system. The rest of the notation is as follows: ns is natural spline; df is the degree of freedom; RH is relative humidity; Time is long-term time trend; Temp is temperature; Holiday is days of national holidays; DOW is the day of the week; Event is the day when the park’s own event was held; Rain is rainy days; Snow is the day it snowed.

By performing Rosner’s test, 110 extreme values were identified and excluded from the park data [11]. Days with the yellow dust were also removed from the data to rule out the effects of the yellow dust alerts (84 days, 2.1%) [12,13]. 

Several sensitivity analyses were performed. First, the threshold was changed from 40 µg/m^3^ to 110 µg/m^3^ to identify whether the result changes according to different thresholds. Secondly, categorical alert results (“Good” or “Normal” versus “Bad” or “Very bad”) which were actually issued were included in the model instead of the indicator variable for the threshold of 80 µg/m^3^. The influence of using the surrogate measure can be identified through this analysis. All the statistical analyses were performed using R software version 3.4.1 [14].

## 3. Results

Table 1 shows the descriptive statistics of the study variables. The daily average PM_10_ concentration for the study period (2014–2019) was 42.0 µg/m^3^, and an average of 24,384 people visited Seoul Children’s Grand Park per day. The daily average PM_10_ concentration and the number of visitors to the park decreased over time. In the case of the control period, which was the five years before the study period, the average PM_10_ concentration was 45.1 µg/m^3^ and 28,283 people visited the park daily. During the study period, 115 out of 2061 days (5.6%) exceeded the daily PM_10_ concentration of 80 µg/m^3,^; 145 out of 1740 days (8.3%) exceeded the threshold in the control period.

The effects of the PM_10_ alert system on the number of park visitors are presented in Table 2. The number of park visitors decreased by 0.881 times (11.9%) when the nationwide PM_10_ alert changed from “Normal” to “Bad” (95% CI: 0.808, 0.960). No additional decrease in slope was observed above the threshold for the “Bad” alert (1.000, 95% CI: 0.997, 1.003). For the period before the nationwide PM_10_ alert was introduced (control period), statistically significant reductions were not observed in the intercept (0.954, 95% CI: 0.872, 1.043) nor the slope (0.997, 95% CI: 0.995, 1.000).

Figure 2 presents the results of a sensitivity analysis that estimated the intercept changes using different thresholds. For the study period, the number of park visitors decreased significantly not only at the threshold of 80 µg/m^3^ but also around 60 µg/m^3^ to 90 µg/m^3^. Still, such reductions in the number of park visitors were not observed in the control period.

Similar results were found when the analysis was performed using categorical alert records. People tended to visit the park about 10% less when the PM_10_ alert was “Bad” or “Very bad” compared to “Good” or “Normal” (Table 3). The associations were statistically significant when people found alert results one day before the afternoon in question (05 P.M., 11 P.M.). 

## 4. Discussion

The present study found evidence that the number of park visitors decreased by 11.8% (RR: 0.881, 95% CI: 0.808, 0.960) when a “Bad” alert was issued compared to when the alert was “Normal.” Such a reduction in visitors was not observed before the nationwide PM_10_ alert was introduced.

Although the PM_10_ alert is one of the major environmental policies operated for over a decade in Seoul, its effectiveness has not been well evaluated in Korea. To the best of the authors’ knowledge, one study estimated the effects of the alert system on the behavior patterns of people in Korea. They investigated the relationship between the PM alert and the number of visitors to tourist attractions such as the royal palaces and the national museum, but could not find associations [8]. Inconsistent results compared to the present study might be due to the difference in the dependent variable. Based on the results of the present study, it can be inferred that the PM_10_ alert system might not affect pre-planned activities, such as attending an exhibition, while effectively intervening in more daily routine activities, such as visiting a park.

Regarding health outcomes, there was another study that estimated the effects of PM_10_ alerts on respiratory diseases [15]. They found evidence that the mobile-based warning services reduced respiratory disease patients by 8.6% (*p*-value < 0.01). Several studies exist outside Korea that estimated the effects of the PM alert system on health outcomes. In Canada, Chen et al. (2018) estimated the effects of air quality alerts on various health outcomes using regression discontinuity analysis and showed that the alert announcement reduced asthma-related emergency visits by 25% (95 CI: 1%, 47%) [16]. Alari et al. (2021) investigated the effects of PM_10_ alerts in Paris using a difference-in-difference design and reported a 0.84 (95% CI: 0.78, 0.93) times reduction in cardiovascular mortality when the “Bad” alarm was issued [17].

Among several causal pathways, encouraging people to refrain from going outside would be a primary pathway for the alert to reduce health burden. Identifying the association between the alert and outdoor activities could secure evidence of whether the alert system effectively protects people’s health. Nevertheless, as people can take other precautionary measures, such as wearing a mask, the health benefit of the policy could be different from the estimates of our study. In future studies, efforts to evaluate the alert effects on various health outcomes would be needed in Korea.

According to a sensitivity analysis, the discontinuity effect at the threshold was not clear-cut at 80 µg/m^3^ but showed statistically significant bands from 60 µg/m^3^ to 90 µg/m^3^. The uncertainty in the threshold might be due to using a surrogate measure in PM_10_ concentration. Because the assignment of alerts is determined by the PM_10_ concentration calculated from a prediction model, the predicted PM_10_ concentration should be used in the segmented regression. Due to the absence of the data, however, the ambient PM_10_ concentration was used in the analysis as a substitute. The error between the two might cause uncertainty in the threshold. For example, even though the actual PM_10_ concentration corresponds to “Normal”, 5.3% of the days were wrongly predicted as “Bad”. The percentage of wrongly predicted days as “Normal” when the actual concentration corresponds to “Bad” was 45% (Appendix A). However, these errors did not appear to affect the magnitude of our results significantly. When the exposure was changed to categorical alerts that were actually issued, the estimated policy effect was similar to the result of our main analysis (RR: 0.883, 95% CI: 0.792, 0.985). 

Our results suggest an important implication for air pollution studies regarding exposure measurement errors. Since it is challenging to measure personal PM exposure, most previous epidemiology studies have used ambient PM concentration as a proxy for individual exposure, assuming that the ambient concentration can effectively represent individual exposure. However, if the PM alert system prevents people from going outside during highly polluted days, the error between the ambient PM concentration and personal exposure is bound to increase. Such an increase in exposure measurement error will introduce bias in effect estimates. In the presence of an effective alert system compared to the absence of an alarm system, the effect of ambient PM would be underestimated because personal exposure to PM would be lower than the ambient PM concentration. In fact, the health effect estimates of PM in Seoul showed temporal heterogeneity, with a decreasing trend in recent years [18,19]. From 2005 to 2009, a 10 µg/m^3^ increase in PM_10_ was associated with a 0.61% (95% CI: 0.34%, 0.89%) increase in non-accidental mortalities, while it decreased to −0.06% (95% CI: −0.38%, 0.25%) in 2011–2015 [16]. Among several possible causes, an increase in exposure measurement error by the PM_10_ alert system could form part of the explanation. Future air pollution epidemiologic studies may have to contrive ways to consider such potential errors. 

Our result also has practical implications regarding public health policies. This study analyzed the effect of the alert system for the period before the introduction of the nationwide alert system and utilized it as a control. Even during the control period, however, a PM_10_ alert was in operation by the local government of Seoul. Although it had the same standards as the national alert system, we could not observe significant signs of effects. The main difference between the two, as mentioned in the introduction, is the difference in public awareness and the degree of publicity of alerts. In the case of environmental policies in which compliance acts as a critical factor, simply introducing an intervention seems to be not enough to achieve its intended goals. Actively publicizing the risk of a harmful agent and systemically disseminating related information would be necessary for improving the effectiveness of environmental public health policies.

There are several strengths and limitations in the study. As the Seoul Children’s Grand Park is a highly accessible, well-managed park that is free of charge, the number of visitors would be a good surrogate for people’s daily outdoor activity patterns. Also, the level of measurement error in outcome is expected to be low as automated counting machines count the visitors. In addition to quantifying the policy effects using the segmented regression, this study further compared the result temporally to ensure that the observed benefit is attributable to the nationwide alert system. 

However, the results of this study may not represent the characteristics of the whole Seoul population. It might represent the behavior patterns of those who live nearby the park or who often visit public parks. Although stratified analysis by visitor characteristics such as age and gender could give a better understanding of the policy, such analyses were not performed due to the absence of the data. Also, this study estimated only the effects of the policy on urban park visits despite there being various outdoor activities other than park visits. Investigating the policy effects on various outdoor activities or expanding the park data to venues other than the Seoul Children’s Grand Park would be needed in future research.

## 5. Conclusions

The present study evaluated the effects of the nationwide PM_10_ alert system on the number of urban park visits in Seoul, Korea. The alert system effectively reduced the number of park visitors on the days when episodic spikes in PM concentration were expected. Actively disseminating alert results and promoting PM risk appear to be related to a more effective alert system. Further efforts investigating the policy effects on various health outcomes should be made to confirm whether the observed behavior change results in a reduction in the public health burden attributable to PM. 

## Figures and Tables

**Figure 1 ijerph-19-15372-f001:**
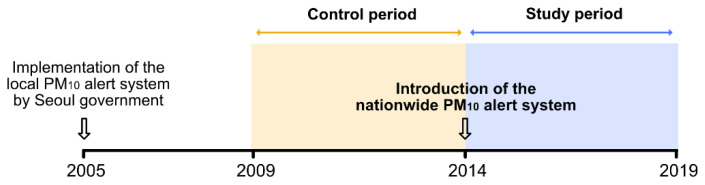
History of PM_10_ alert systems in Seoul, Korea.

**Figure 2 ijerph-19-15372-f002:**
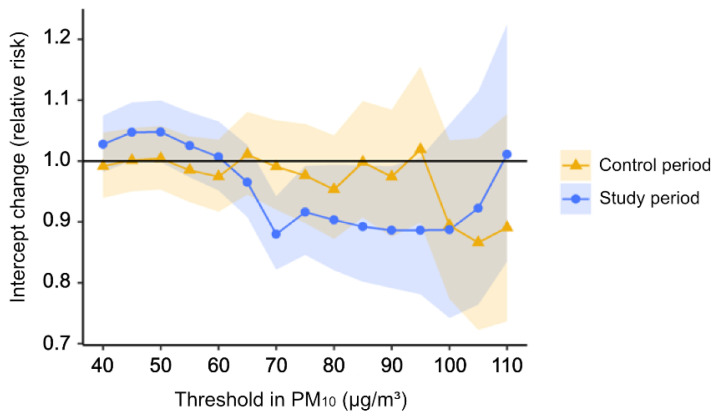
Results of sensitivity analysis by different thresholds of PM_10_.

**Table 1 ijerph-19-15372-t001:** Daily averages and standard deviations for the study variables in Seoul, Korea.

	PM_10_ (µg/m^3^)	Park Visitors (n)	Temperature (°C)	Relative Humidity (%)
Study period ^a^	42.0 (21.9)	21,091 (14,424)	13.6 (10.7)	59.4 (14.8)
Control period ^b^	45.1 (24.5)	28,283 (18,334)	12.1 (11.3)	60.3 (15.3)

^a^ 31 July 2014–31 December 2019; ^b^ 1 January 2009–6 February 2014.

**Table 2 ijerph-19-15372-t002:** Effects of the PM_10_ alert system on the number of visitors to an urban park in Seoul, Korea.

	Relative Risk (95% CI)	
	Control Period	Study Period
Intercept change at 80 µg/m^3^	0.954 (0.872, 1.043)	0.881 (0.808, 0.960)
Slope change after 80 µg/m^3^	0.997 (0.995, 1.000)	1.000 (0.997, 1.003)

**Table 3 ijerph-19-15372-t003:** The relative risk of visiting an urban park when the PM_10_ alert was “Bad” or “Very bad” compared to “Normal” or “Good” in Seoul, Korea, 2014–2019.

Prediction	Alert Time	Relative Risk (95% CI)
Today	05 A.M.	0.947 (0.851, 1.054)
	11 A.M.	0.913 (0.816, 1.022)
	05 P.M.	0.894 (0.794, 1.008)
	11 P.M.	0.891 (0.780, 1.017)
Tomorrow	05 A.M.	0.968 (0.853, 1.097)
	11 A.M.	0.900 (0.785, 1.032)
	05 P.M.	0.883 (0.792, 0.985)
	11 P.M.	0.894 (0.802, 0.996)

## Data Availability

Data in this study were from Seoul Facility Corporation, National Institute of Environmental Research and Korea Meteorological Administration.

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
