# Peer review of "The Effects of Particulate Matter Alert on Urban Park Visitation in Seoul, Korea: Using Segmented Regression"

_ijerph, 2022, doi:10.3390/ijerph192215372_

Round 1

Reviewer 1 Report

I have enjoyed reading this manuscript. I have the followings minor comments/suggestions to improve the overall quality of the manuscript.
2.2 The reference for equation used is not given. Please check

Reference number 10 is not present in the material and method section.

Results can also be evaluated on age category for a better understanding of public awareness. Because in parks more children or women must be present rather than younger people, also should highlight sensitive age groups. 

Summarize the effect of PM on the number of visitors on working days’ vs weekends (if possible).

Writing manuscripts with high-quality English language is important from the reader's point of view. Therefore, it is suggested to have the article checked for English language and grammatical errors. 

The use of capital letters in the subject and subtopic is inconsistent.

Figure 2. Axis labels are too small to be visible.

Please use PM10 ­­throughout i.e. 10 in subscript.

Conclusions need to be clear and reflect the importance and highlights of the research.

The discussion section is very well covered.

Author Response

Thank you for taking the time to review our manuscript. We made an extensive revision of the manuscript, including English and grammar. Your detailed comments have been a great help in revising the manuscript. Below are responses to comments.

Comment 1: The reference for equation used is not given. Please check

Response: We added two references regarding the equation. The first is for the time-series regression (reference number 10), and the second is for the segmented regression (reference number 9).

Comment 2: Reference number 10 is not present in the material and method section.

Response: Reference number 10 was about the Rosner test used to identify extreme data points and was in line 142. It is now located in 135, as a reference number of 11.

Comment 3: Results can also be evaluated on age category for a better understanding of public awareness. Because in parks more children or women must be present rather than younger people, also should highlight sensitive age groups.

Response: We agree that it would be better if we could perform a stratified analysis according to the characteristics of the visitors. However, we could only obtain the data for the number of daily visitors to the park, not for the visitor characteristics. It remains one of our limitations, and the related contents are specified in limitation lines 263-267.

Comment 4: Summarize the effect of PM on the number of visitors on working days' vs weekends (if possible).

Response: This is also an important point. When we analyzed the data only for working days, a similar result was found with our main result: a 9% decrease in the visitors on the days with the "bad" alert was issued (relative risk 0.9102, 95% confidence interval: 0.8278, 1.0008). We could not analyze the effect on weekends because of the limited number of days. Instead, we adjusted the day of the week in the main model to consider possible confounding effects from the different numbers of visitors on weekends.

Comment 5: Writing manuscripts with high-quality English language is important from the reader's point of view. Therefore, it is suggested to have the article checked for English language and grammatical errors. 

Response: We revised the manuscript thoroughly.

Comment 6: The use of capital letters in the subject and subtopic is inconsistent.

Response: We corrected it according to the journal guideline.

Comment 7: Figure 2. Axis labels are too small to be visible.

Response: The font size of the axis label has been increased.

Comment 8: Please use PM10 ­­throughout *i.e*. 10 in subscript.

Response: We corrected them as a subscript.

Comment 9: Conclusions need to be clear and reflect the importance and highlights of the research.

Response: We rewrote the conclusion. Thank you again for your great comments.

Reviewer 2 Report

The idea is interesting, but i think it can be improved, particularly with the supplementary over the survey of willing not enter the park.

1. I wonder how many people were alerted by the alert system in the park. Only when the people go to the park would be alerted, but not by the system before they go to the park. That kind of system would be more meaningful to the public health.

2. Only one park may not be enough illustrate the effect of the alert system. Besides, there are more event days after the alert system being implemented. 

3. The author may need to provide significant sample of survey over the reasons for people stopping going to the park.

4. Better choice is that the government need to shut down the park when the PM concentration is high. That would be more effective.

Author Response

Thank you for taking the time to review our manuscript. Through your comments, we could figure out what our paper was missing and could make up for the shortcomings. The manuscript has been revised extensively, including English corrections. Please consider the following responses:

In the previous manuscript, the description of our data seemed insufficient. To explain further, the number of park visitors was not measured through a survey. We obtained public open data provided by the park management facility, the Seoul Facility Corporation. In the case of the Seoul Children's Grand Park, visitors can only enter the park through designated entrances, and automated visitor counting machines installed at all entrances measure the number of visitors to the park. Therefore, the subjects of our study are all those who visit the park, not a sample of visitors.

Also, regarding the PM10 alerts, the government predicts the PM10 concentration one or two days before and distributes warnings through various channels, including television, the internet, etc. The alert is not issued within the park but is available in various sources in our daily life.

We revised the description of the park visit data and PM alert system thoroughly in the method section.

Below are the response to the comments.

Comment 1: I wonder how many people were alerted by the alert system in the park. Only when the people go to the park would be alerted, but not by the system before they go to the park. That kind of system would be more meaningful to the public health.

Response: As explained above, the alerts are not issued at the park but issued one or two days before they go to the park, and people can access the information through various channels. According to a study that investigated people's perception of the alerts, 738 out of 1,111 people surveyed (66%) answered that to have searched for the alert to take precautionary measures.

Comment 2: Only one park may not be enough illustrate the effect of the alert system. Besides, there are more event days after the alert system being implemented.

Response: The Seoul Children's Grand Park is a unique park that measures all the number of visitors using automated counting machines even though admission is free. We agree that analyzing the policy effects using various park data would be better. However, we could not include other park data because of the data availability. We added comments regarding the limitation of data availability in lines 267-271.

Nevertheless, we believe that the Seoul Children's Grand Park is large enough to estimate the behavior patterns of the surrounding residents, having a daily average number of visitors of more than 20,000 persons.

Comment 3: The author may need to provide significant sample of survey over the reasons for people stopping going to the park.

Response: As explained above, our study is not a survey research but a study that used secondary data that measured all the visitors to the park.

Comment 4: Better choice is that the government need to shut down the park when the PM concentration is high. That would be more effective.

Response: The comment could be one of the various political options to manage PM risk by the government. All public health policies should be established based on valid scientific evidence, and we believe our manuscript can provide such evidence. Thank you for reviewing our manuscript. 

Reviewer 3 Report

1. Although many authors use the first person in their articles, scientific papers should be impersonal. Please correct the paper in the third person.

2. The time frame needs to be updated. I suggest revising the article to update the data until 2021. Otherwise, the timeliness of publication is harmed.

Author Response

Thank you for taking the time to review our manuscript. We extensively revised the manuscript, referring to the comments, including English and grammar. Please consider the following responses to the comments.

Comment 1: Although many authors use the first person in their articles, scientific papers should be impersonal. Please correct the paper in the third person.

Response: We changed all the first-person expressions to impersonal sentences.

Comment 2: The time frame needs to be updated. I suggest revising the article to update the data until 2021. Otherwise, the timeliness of publication is harmed.

Response: Thank you for a great comment. We agree that extended time could provide more reliable evidence. However, we set the study period until 2019 because of the pandemic outbreak (Covid-19). After 2020, the outdoor activity pattern of people changed completely because of the lockdowns. To exclude the confounding effects of the pandemic from our analysis, we could not extend the study period after 2019.

Reviewer 4 Report

My comments are mainly as follows:

·         1. Some spaces exist between units and numbers, some not. such as the “80μg/m3” in the abstract should be changed to “80 μg/m3”. Please go through it carefully.

·         2. What are the health problems caused as mentioned in the abstract?

·         3. The mechanism of the alert system is not clarified in the manuscript.

·         4. Figure 1 are unclear and the details are needed to fill..

·         5. Why the PM alert system mentioned in the introduction is not significantly associated with outdoor activities? Please give the reasons.

·         6. National Institute of Environmental Research (NIER) and Korea Meteorogical Administration in materials and methods should preferably be added to the web site of the data source.

·         7. The description of figure 2 is not clear enough, and the description of image is less.

·         8. In addition to the PM alert system, are there other factors that affect people's exposure to high PM10 concentrations? Is the improvement of population health related to other influencing factors?

Author Response

Thank you for taking the time to review our manuscript. We made an extensive revision of the manuscript, including English and grammar. Your detailed comments have been a great help in revising the manuscript. Below are responses to comments.

Comment 1: Some spaces exist between units and numbers, some not. such as the "80μg/m3" in the abstract should be changed to "80 μg/m3". Please go through it carefully.

Response: We examined all the errors in the manuscript and corrected them.

Comment 2: What are the health problems caused as mentioned in the abstract?

Response: We added descriptions of the health effects of PM in lines 32-36.

Comment 3: The mechanism of the alert system is not clarified in the manuscript.

Response: According to the reviewer's comment, we revised the method thoroughly. The mechanism of the alert system is now described in lines 79-83.

Comment 4: Figure 1 are unclear and the details are needed to fill.

Response: Figure 1 represents the history of PM10 alert system in Korea. We changed the wording in the graph more clearly. The descriptions for Figure 1 are now in lines 71-78.

Comment 5: Why the PM alert system mentioned in the introduction is not significantly associated with outdoor activities? Please give the reasons.

Response: We believe that Shin et al. (2022) could not find significant effects of the PM alert system because they estimated the policy effect on the number of visitors to tourist attractions. Visiting the national museum or a royal palace is likely to be pre-planned and accompany a party. On the contrary, our finding is about more casual daily activity, visiting a public park. The difference between the previous study and ours are written in lines 186-192.

Shin, Y., et al., Effectiveness of Particulate Matter Forecasting and Warning Systems within Urban Areas. Sustainability, 2022. 14(9): p. 5394.

Comment 6: National Institute of Environmental Research (NIER) and Korea Meteorogical Administration in materials and methods should preferably be added to the web site of the data source.

Response: We added website links as suggested.

Comment 7: The description of figure 2 is not clear enough, and the description of image is less.

Response: Figure 2 is about the sensitivity analysis by fitting different thresholds. We changed axis labels and described the contents in detail in lines 167-171

Comment 8: In addition to the PM alert system, are there other factors that affect people's exposure to high PM10 concentrations? Is the improvement of population health related to other influencing factors?

Response: Thank you for a great comment.

Among various policies implemented in Korea, the yellow dust alert can affect people's exposure at high PM concentrations. Therefore, we excluded the days with the yellow dust from our data (lines 135-136).

Also, we used segmented regression to exclude the confounding effects from other policies. The policy effect was evaluated by estimating discontinuity effects at 80 µg/m3 of PM10 concentration. Improving population health would reduce the dose-response line's overall slope (relative risk), not causing the discontinuity effects at the threshold. Considering these points, we believe that the degree of confounding by other policies might be low.

Round 2

Reviewer 3 Report

I stand by my previous decision.

The time frame is too old, and the study should be implemented even with the authors' justification for post-pandemic changes.

Success in future submissions.

Author Response

We appreciate your review. 
We will develop future research considering the time frame of the study.

Reviewer 4 Report

I think the authors have answered the comments properly. However, there are some editorial errors, e.g. the notes writen in Korean should not appared in the revised version.

Author Response

We deleted notes written in Korean. 

Thank you for the constructive review.